# PREDICTION OF PROTEIN-PROTEIN CONTACTS WITH STRUCTURE-AWARE SINGLE-SEQUENCE PROTEIN LANGUAGE MODELS

## ABSTRACT

Accurate prediction of the interface residue-residue contacts between interacting proteins is valuable for determining the structure and function of protein complexes. Recent deep learning methods have drastically improved the accuracy of predicting the interface contacts of protein complexes. However, existing methods rely on Multiple Sequence Alignments (MSA) features which pose limitations on prediction accuracy, speed, and computational efficiency. Here, we propose a transformer-powered deep learning method to predict the inter-protein residue-residue contacts based on both single-sequence and structure-aware protein language models (PLM), called DeepSSInter. Utilizing the intra-protein distance and graph representations and the ESM2 and SaProt protein language models, we are able to generate the structure-aware features for the protein receptor, ligand, and complex. These structure-aware features are passed into the Resnet Inception module and the Triangle-aware module to effectively produce the predicted inter-protein contact map. Extensive experiments on both homo- and hetero-dimeric complexes show that our DeepSSInter model significantly improves the performance compared to previous state-of-the-art methods.

## 1 INTRODUCTION

Understanding the interactions between proteins is fundamental to deciphering the molecular mechanisms underlying cellular processes (Hu et al., 2021; Wu et al., 2024b). Accurate prediction of the interface residue-residue contacts of protein-protein interactions (PPI) allows for the determination of the resulting protein complex structure (see Figure 1 for an example) (Gao et al, 2024), having significant implications for understanding the protein complex's biological function, increasing the efficiency for drug discovery, and saving time and resources for experimental methods (Lin et al., 2024a). However, current methods still lack accuracy and efficiency when predicting the interface contacts of protein complexes. Computational methods include more traditional methods such as docking simulation (Yan et al., 2020; Yu et al., 2024; Wu et al., 2024a; Honorato et al., 2024) and coevolutionary analysis (Ovchinnikov et al., 2014), as well as more recent deep learning methods (Liu & Gong, 2019; Zeng et al., 2018; Adhikari et al., 2018; Quadir et al., 2021b; Roy et al., 2022; Quadir et al., 2021a; Yan and Huang, 2021) that can capture the complex patterns in protein sequences and structures. However, the former lacks in scalability and generalizability across different types of protein complexes, especially heterodimers; while the latter still lacks in accuracy and ability to fully leverage the structural context of proteins, which is crucial for accurate contact prediction (Lin et al., 2024a).

Recent advances in deep learning have led to more sophisticated models for inter-protein contact predictions such as DeepInter(Lin et al., 2023), DeepHomo2.0(Lin et al., 2022), GLINTER(Xie & Xu, 2022), and CDPred (Guo et al., 2022), or direct prediction of protein complex structures like RosettaFold (Baek et al., 2021) and AlphaFold-Multimer (AFM) (Evans et al., 2021). However, all current methods rely on input of the multiple sequence alignment information of proteins. Use of MSA information can be beneficial, but there exist several drawbacks and challenges associated with integrating MSA data into deep learning models. First, usage of MSA information brings high computational complexity. MSA data can be very large, especially for long sequences and large sequence databases. Preprocessing of MSA data, including generating the alignment and converting

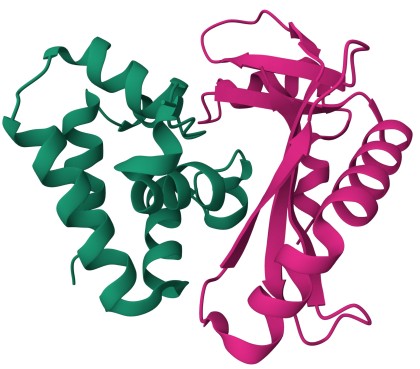

Figure 1: An example of the protein complex structure between CdiA-CT/CdiI from Y. kristensenii 33638 (PDB code: 5E3E).

it into a usable format is time-consuming. Second, in order for the deep learning model to effectively capture the features necessary for interface contact prediction, the input MSA information needs to be of high quality. However, MSAs may contain gaps or insertion/deletions, which could lead to noise in the model, leading to inaccurate predictions. Sequences with few homologs or from underrepresented groups may also result in poorly constructed alignments or not exist altogether, causing poor performance and lack of generalizability (Lin et al., 2023). Third, it requires to pair the MSA of each monomer proteins to construct the paired MSA for extracting the coevolutionary features at the interface, which is still a great challenge in the field of protein structure prediction (Bitbol et al., 2016; Szurmant et al., 2018; Gueudré et al., 2016; Ovchinnikov et al., 2014; Zeng et al., 2018; Bryant et al., 2022; Chen et al., 2023). Thus, improved methods are needed to overcome these limitations.

In contrast to MSA-reliant methods, single sequence-based methods have also been developed to predict protein monomer structures (Wang et al., 2022; Chowdhury et al., 2022; Fang et al., 2023; Jing et al., 2024; Lin et al., 2024b). Single sequence-based methods utilize evolutionary information extracted from protein language models (PLM) to effectively capture features. Unlike MSA which requires high computational resources, single sequence-based methods are less computationally intensive and are faster in speed. In addition, single sequence-based methods do not rely on the availability of homologous sequences. This allows for prediction and design of novel or engineered protein where MSA information is not applicable (Chowdhury et al., 2022; Watson et al., 2023; Ren et al., 2024; Shi et al., 2022). Furthermore, single sequence-based methods may be better at capturing the structural and interaction properties of dynamic or disordered regions (Jing et al., 2024). These regions are often poorly aligned in MSAs, making single sequence-based methods favorable. The advantages of single sequence-based methods over MSA-reliant methods make them more reliable, scalable, and robust. However, there still exists a lack of well-performing single sequence-based methods for the interface contact prediction of protein complexes.

To address this need, we propose DeepSSInter, a single sequence-based deep learning model for the interface contact prediction of protein complexes. We utilize two protein language models (PLM) to effectively capture evolutionary and structural patterns of input monomer proteins. Specifically, one of the PLMs is ESM2, which takes in the monomer sequences of the proteins (Lin et al., 2024b), and the other is SaProt, which takes in the structure-aware sequence of the proteins generated with their 3D structure information (Su et al., 2023). Due to the input of structure-aware sequences into the latter PLM, the resulting representations and attentions contain structural information of individual proteins. Our model leverages this information to provide faster and more accurate predictions of interface contacts. To validate the effectiveness of our method, we comprehensively evaluated our model on diverse data sets of homodimeric and heterodimeric protein complexes. It is shown that our model outperforms previous state-of-the-art methods, especially when predicting challenging heterodimer complexes, establishing the effectiveness of our single-sequence and structure-aware protein language model.

The main contributions of our model are summarized as follows:

- We propose a transformer-based deep learning method for single-sequence inter-protein contact prediction by effectively integrating both single-sequence and structure-aware protein language models.

- The model does not rely on the input of multiple sequence alignment (MSA) alignment, allowing the model to be more computationally efficient with better or similar accuracy compared to MSA-reliant methods.

- The model is powered by the ResNet-Inception module, which can efficiently capture the long-range interaction between pairs of residues, and a geometric triangle-aware module, which is able to consider the many-body effect in residue-residue interactions.

- Our method utilizes intra-protein information, graph representation, single-sequence, and structure-aware features that can effectively capture evolutionary and structural patterns of input individual proteins. As a result, the performance of our method significantly surpasses stat-of-the-art prediction methods.

## 2 RELATED WORK

Currently, state-of-the-art methods for predicting the inter-protein residue-residue contacts of protein complexes all require the MSA as input.

### 2.1 DEEPHOMO2.0

DeepHomo2.0 (Lin et al., 2022) predicts the inter-protein contact probabilities of homodimeric complexes by combining sequential 1D features and pairwise 2D features, passing them through convolutional neural networks. The model achieves high accuracy by integrating evolutionary information, residue-residue distance maps, and transformer-derived context. However, DeepHomo2.0 has limited generalizability to other types of protein-protein interactions such as heterodimers because the model is specifically designed for homodimeric protein complexes.

### 2.2 GLINTER

GLINTER (Xie & Xu, 2022) predicts the interface contact probabilities by using graph representations and MSA features through a graph convolutional network with an attention mechanism and ResNet layers. GLINTER improves the accuracy and robustness in interface contact prediction of protein complexes, but still faces challenges with computational complexity and protein dimer precision.

### 2.3 CDPRED

CDPred (Guo et al., 2022) predicts inter-chain distance maps of protein complexes by passing features into a model consisting of a deep residual network, a channel-wise attention mechanism, and a spatial-wise attention mechanism. Despite its effectiveness, especially for homodimers, it faces challenges on heterodimeric complexes or the cases with shallow MSAs.

### 2.4 DEEPINTER

DeepInter (Lin et al., 2023) predicts interface contact probabilities of protein complexes by incorporating a ResNet-Inception module and triangle-aware mechanism that can capture geometric consistency and long-range interactions. This allows DeepInter to provide more accurate and robust contact predictions compared to existing methods. However, it lacks in still lacks in accuracy for heterodimers and relies on high-quality MSA information.

# 3 MODEL ARCHITECTURE

## 3.1 OVERVIEW OF DEEPSSINTER

Figure 2 shows the overall architecture of the DeepSSInter network and the flow of data through the network. The architecture consists of four main components: 1) a geometric graph transformer module (Morehead et al., 2021) consisting of a graph neural network that generates protein sequence representations, 2) two protein-language models (ESM2 (Lin et al., 2024b) and SaProt (Su et al., 2023)) to generate protein sequence-aware and structure-aware sequence and attention representations, 3) a ResNet-Inception module (Lin et al., 2023), and 4) a triangle-aware module consisting of the triangle update, triangle self-attention, and transition layers (Lin et al., 2023). Graph representations of the individual proteins are passed into the geometric transformer module to generate sequence representations of the proteins. The sequences of the individual proteins are passed into the ESM2 and SaProt models to generate sequence and attention representations, for which those of SaProt are structure-aware. In addition, the linked sequences of individual proteins are also passed into ESM2 and SaProt to generate attention representations associated with the interface. Then, the sequence representations, attention representations, and distance features, obtained by applying a radial basis function on the locations of the residues within each protein, are respectively concatenated into sequence-aware features and passed into the ResNet-Inception module. The data is finally passed into the triangle-aware module from the ResNet-Inception module. At the prediction time, only one (for a homodimer case) or two (for a heterodimer case) monomer structures are needed as input for the model. By default, DeepSSInter does crop the monomer structure during the inference. However, for a very long protein, users may use a sliding window strategy (Appendix F). The output is a contact map consisting of the pairwise probabilities between the amino acids of two proteins.

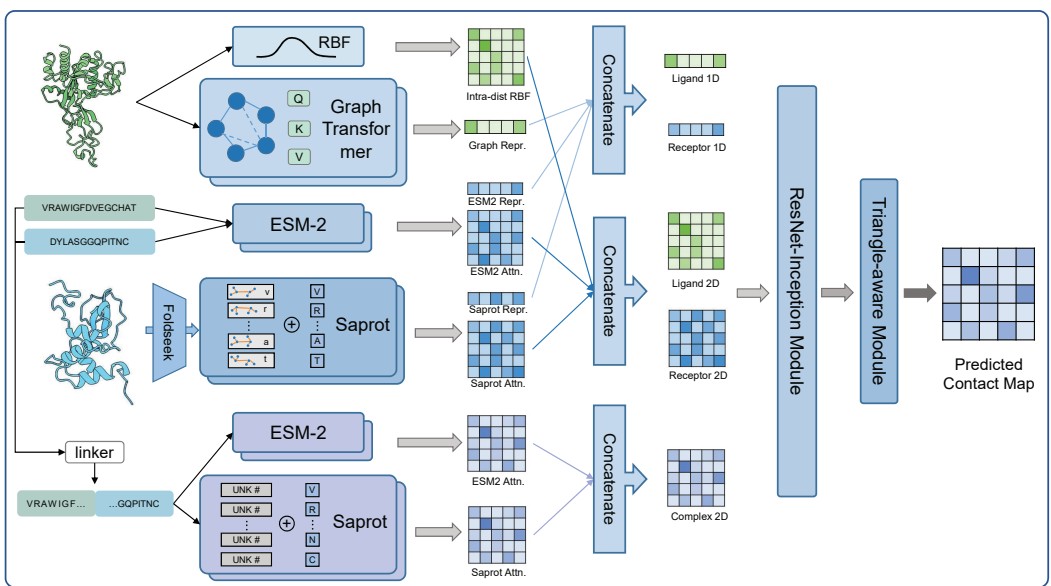

Figure 2: The workflow of the DeepSSInter network.

## 3.2 NETWORK ARCHITECTURE AND PARAMETERS

The model takes in the graph representations, intra-protein distance matrices, sequences, and structure-aware sequences (generated by foldseek (van Kempen et al., 2024) for SaProt (Su et al., 2023)) of two proteins as input (Figure 2). The graph representations of two individual proteins are passed through the geometric transformer to obtain two representations of dimensions $L_A \times 128$ and $L_B \times 128$ respectively, where $L_A$ is the length of protein A and $L_B$ is the length of protein B.

The intra-protein distance feature of individual proteins is represented by the Gaussian radial basis function (RBF), which is calculated as follows.

$$\varphi(d) = e^{-(\frac{d-d_\mu}{\sigma})^2}$$

where $d$ is the intra-protein distance. $d_\mu$ is a hyperparameter representing the centers of 64 evenly spaced Gaussian RBFs between 2 to 22 Å with standard deviation $\sigma = 0.3125$. The results are two distance matrices of dimensions $L_A \times L_A \times 64$ and $L_B \times L_B \times 64$ for protein A and protein B respectively.

The amino acid sequences of the two proteins are passed into the ESM2 protein language model (specifically the esm2_t33_650M_UR50D model with 33 layers and 650 million parameters) to generate representation matrices of dimensions $L_A \times 1280$ and $L_B \times 1280$, respectively, and attention matrices of dimensions $L_A \times L_A \times 660$ and $L_B \times L_B \times 660$, respectively, for protein A and protein B. We also concatenate sequence A and sequence B and pass the resulting complex sequence into the ESM2 model to get the representation and attention matrices of dimensions $(L_A + L_B) \times 1280$ and $(L_A + L_B) \times (L_A + L_B) \times 660$ for the protein complex.

The structure-aware sequences generated with the foldseek algorithm of two proteins are passed into the SaProt protein language model (specifically the SaProt_650M_AF2 model trained with a dataset of 40 million AlphaFold2 structures (Varadi et al., 2024)) to generate sequence-aware representation and attention matrices. For protein A and protein B, the dimensions of the representation matrices are $L_A \times 1280$ and $L_B \times 1280$, respectively, and the attention matrices are of dimensions $L_A \times L_A \times 660$ and $L_B \times L_B \times 660$, respectively. Similar to before, we also concatenate structure-aware sequence A and structure-aware sequence B and pass the resulting complex sequence into the SaProt model to get the structure-aware representation and attention matrix of dimensions $(L_A + L_B) \times 1280$ and $(L_A + L_B) \times (L_A + L_B) \times 660$ respectively for the protein complex.

We then concatenate the geometric transformer representations, ESM2 representations, and SaProt representations of protein A and protein B to obtain 1D features for the two proteins. We also concatenate the ESM2 attention matrices, SaProt attention matrices, and RBF distance matrices of protein A and protein B to obtain the 2D features for the two proteins. Finally, we concatenate the ESM2 attention matrix and the SaProt attention matrix of the protein complex (protein A + protein B) to obtain 2D features for the protein complex. The two 1D features of protein A and protein B, the two 2D features of protein A and protein B, and the 2D features of the protein complex are passed through linear layers performing dimensionality reduction to prevent overflow of GPU memory. These features are then passed into the Resnet-Inception module. The ResNet-Inception module outputs the receptor (protein A), ligand (protein B), and complex features (protein complex) of dimensions $L_A \times L_A \times d$, $L_B \times L_B \times d$, and $L_A \times L_B \times d$, respectively, where $d$ is a hyperparameter set as 64. The receptor, ligand, and complex 2D structures are passed into the Triangle-aware module. The model finally outputs the pairwise probabilities that the residue-residue contacts exist between protein A and protein B.

### 3.3 IMPLEMENTATION OF TRAINING

DeepSSInter uses Focal Loss as the loss function for training, the same loss function as that used for DeepInter (Lin et al., 2023). Compared to other classification loss functions such as standard Cross-Entropy, Focal Loss is able to address the issue of class imbalance. In the case of interface contact prediction, there is an extreme class imbalance between non-contacts and contacts, with the number of non-contacts greatly outnumbering the number of contacts. Focal loss helps the model focus on the minority class, the true contacts in this case. Focal loss also makes the model focus more on difficult, misclassified examples, improving the model's ability to generalize on challenging test cases. Focal loss is defined by:

$$FocalLoss(p_t) = -\alpha_t(1 - p_t)^\gamma \log(p_t)$$

$$p_t = \begin{cases} p & \text{if } y = 1 \\ 1-p & \text{otherwise,} \end{cases}$$

where $y \in 0, 1$ is the ground truth label. We use parameters of $\alpha_t = 0.25$ and $\gamma = 1.5$ (Lin et al., 2017).

Our model is trained using PyTorch Lightning on one A100 GPU with 40G memory. We trained the model with a learning rate of 0.001 and a weight decay of 0.01. Due to GPU memory limitations of each module, especially the triangle self-attention module, we set a maximum sequence length of 320 for each input protein. If the input protein has a length greater than 320, we use a window of size 320 to scan the labels and find the windows that have the maximum number of inter-protein contacts. From the windows with the most contacts, we randomly select a window and crop the protein sequence and input features to match the window. For the ground truth labels of the interface contacts, we consider two amino acids with a distance of $< 8.0$ Å among their heavy atom pairs from two proteins in the complex to be an interface contact.

## 4 Experiments

### 4.1 Datasets

To train and test our model, we use the data sets of non-redundant protein dimeric complexes from DeepInter, which include 4100 homodimers and 2076 heterodimers (Lin et al., 2023). All the structures were downloaded from the Protein Data Bank (PDB; http://www.rcsb.org/pdb/) (Berman et al., 2000) and subject to manual curation. In this study, we apply the geometric graph transformer to process the protein structure and extract the structure representation and utilize the ESM-2 to obtain sequence representation. Since some residues are missed in the experiment, and to avoid the large gap between the full sequence and structures, we have removed some dimers from the datasets used by DeepInter. For training, the dataset consists of 3376 homodimeric protein complexes and 1853 heterodimeric protein complexes. We use a validation set consisting of 287 homodimeric protein complexes and 95 heterodimeric protein complexes. For testing, we use two test sets of 289 homodimeric protein complexes, which is named Homodimer289, and 99 heterodimeric protein complexes, which is called Heterodimer99.

### 4.2 Evaluation on Homodimeric Complexes

We evaluated our model on the 289 homodimeric protein complexes from the Homodimer289 test set by measuring the mean top-k precision with k = 1, 10, 25, 50, L/10, L/5, L where L is the length of the complex. We compared these metrics with five other state-of-the-art methods: Deep-Inter, CDPred, DeepHomo2.0, GLINTER, and DeepHomo. The top-k precision is defined as the percentage of correct contacts among the top k predicted contacts with highest probability. It can be seen from Table 1 that DeepSSInter obtains high precisions of 83.4%, 81.6%, 80.7%, 79.8%, 80.7%, 79.8%, and 75.0% for top 1, top 10, top 25, top 50, top L/10, top L/5, and top L predicted contacts, respectively, with experimental sequences and structures as input into the model. DeepSS-Inter achieves the highest top-k precisions for all seven top-k precisions among the six methods. In addition, DeepSSInter also obtains the best performance in terms of F1-score and AUC (Table 5).

Table 1: Comparison of the precisions (%) of DeepSSInter and five other methods on the Homodimer289 test set considering the top 1, 10, 25, 50, L/10, L/5, and L predicted contacts with the experimental sequences and structures as input. The data of the other methods are taken from the literature (Lin et al., 2023).

| Method | Top 1 | Top 10 | Top 25 | Top 50 | Top L/10 | Top L/5 | Top L |
|---|---|---|---|---|---|---|---|
| **DeepSSInter** | **83.4** | **81.6** | **80.7** | **79.8** | **80.7** | **79.8** | **75.0** |
| DeepInter | 80.3 | 78.5 | 77.8 | 77.0 | 77.9 | 77.1 | 71.3 |
| CDPred | 74.0 | 71.9 | 69.8 | 67.9 | 69.7 | 67.9 | 58.4 |
| DeepHomo2.0 | 74.0 | 71.7 | 69.5 | 67.0 | 69.6 | 67.2 | 54.7 |
| GLINTER | 68.9 | 64.1 | 60.2 | 56.5 | 61.2 | 57.4 | 43.4 |
| DeepHomo | 61.6 | 57.3 | 54.0 | 50.4 | 54.9 | 52.0 | 39.1 |

In particular, compared to DeepInter, the best-performing method out of the other five methods, DeepSSInter achieves a 3~4% improvement for each top-k precision result. DeepSSInter uses geometric transformer representations and structure-aware PLM representations and attentions, while

DeepInter uses MSA, intra-protein distance, and coevolution information as features. The improvement in precision indicates that the geometric transformer and structure-aware PLM representations and attentions do a better job at capturing the relevant features important for predicting interface contacts.

Compared with the other four methods (excluding DeepInter), DeepSSInter also achieves improvements of 9.4∼16.6%, 9.4∼20.3%, 14.5∼31.6%, and 21.8∼35.9% compared to CDPred, Deep-Homo2.0, GLINTER, and DeepHomo, respectively. This indicates the superiority of DeepSSInter and the effectiveness of DeepSSInter in accurately predicting the interface contacts of homodimeric complexes.

Not all proteins have existing experimental structures available. Therefore, we further tested our model using the predicted monomer protein structures by AlphaFold2 (Jumber et al., 2021) as input to investigate the robustness of our method. We also used the Homodimer289 test set, but instead of inputting the experimental protein sequences and structure, we input the AlphaFold2-predicted structures generated using only the sequences of the proteins. The resulting precisions are shown in Table 2. As we can see, DeepSSInter achieves top-k precisions of 71.6, 69.2, 68.9, 68.2, 68.6, 68.1, and 62.9% for k = 1, 10, 25, 50, L/10, L/5, and L predicted contacts, respectively. In addition, DeepSSInter also achieves the overall best performance when considering both F1-score and AUC (Table 6). Compared with the precisions obtained by inputting experimental sequences and structures, the precisions obtained by inputting AlphaFold2-predicted structures are lower in general. This decrease in performance may be due to the fact that the model is trained on a training set consisting of experimental sequences and structures, thus it would perform better in predicting protein interface contacts for experimental sequences and structures. Nevertheless, DeepSSInter still achieves the highest top-k precisions for every k value among all six methods. This demonstrates the robustness of DeepSSInter, which is able to achieve high precisions even when the input data is changed to AlphaFold2-predicted structures instead of experimental structures.

In addition to inter-protein contact prediction approaches, methods have also been developed for direct prediction of protein complex structures using MSA like AlphaFold-Multimer (Evans et al., 2021), PLM like ESM-Fold (Lin et al., 2024b) and Uni-Fold MuSSe (Zhu et al., 2023), and docking like HDOCK (Yan et al., 2020). Therefore, we have also evaluated these complex structure prediction methods. As shown in Table 2, DeepSSInter also outperforms three typical methods including HDOCKlite (ab initio docking version of HDOCK), ESM-FOLD, and AlphaFold-Multimer (AFM) w/o MSA in predicting inter-protein contacts of homodimers.

Table 2: Comparison of the precisions (%) of DeepSSInter and five other methods on the Homodimer289 test set considering the top 1, 10, 25, 50, L/10, L/5, and L predicted contacts with the full sequences and AlphaFold2-predicted structures as input. The data of the other methods are taken from the literature (Lin et al., 2023).

| Method | Top 1 | Top 10 | Top 25 | Top 50 | Top L/10 | Top L/5 | Top L |
|---|---|---|---|---|---|---|---|
| **DeepSSInter** | **71.6** | **69.2** | **68.9** | **68.2** | **68.6** | **68.1** | **62.9** |
| DeepInter | 69.2 | 66.9 | 66.8 | 65.7 | 66.7 | 65.9 | 59.0 |
| CDPred | 68.5 | 67.5 | 66.6 | 64.5 | 66.8 | 64.7 | 54.6 |
| DeepHomo2.0 | 62.6 | 62.1 | 60.4 | 58.1 | 60.8 | 58.2 | 46.9 |
| GLINTER | 60.6 | 56.5 | 54.1 | 50.9 | 54.5 | 51.5 | 39.1 |
| DeepHomo | 55.7 | 50.1 | 46.8 | 44.2 | 48.0 | 44.8 | 33.7 |
| HDOCKlite | 63.1 | 62.7 | 62.4 | 62.2 | 62.7 | 62.4 | 58.9 |
| ESMFold | 59.7 | 60.1 | 60.0 | 59.8 | 60.1 | 60.1 | 56.7 |
| AFM w/o MSA | 12.3 | 12.7 | 12.4 | 12.7 | 12.5 | 12.7 | 11.6 |

## 4.3 EVALUATION ON HETERODIMERIC COMPLEXES

We further evaluated our model on the more difficult heterodimeric protein complexes. Compared to homodimeric complexes, prediction methods struggle to accurately predict the structure of heterodimeric complexes. We tested our model on the Heterodimer99 test set consisting of 99 heterodimeric protein complexes and measured the top-k precisions with k = 1, 10, 25, 50, L/10, L/5, L

where L is the length of the protein complex. We compared these metrics with three other methods: DeepInter, CDPred, and GLINTER. We can see from Table 3 that DeepSSInter achieves top-k precisions of 59.6%, 50.0%, 48.8%, 45.7%, 50.8%, 48.8%, and 41.9% for k values of 1, 10, 25, 50, L/10, L/5, and L, respectively, for experimental sequences and structures as input into the model. DeepSSInter also obtains significantly higher top-k precisions for all seven top-k precisions among the four methods. In addition, DeepSSInter also achieves the overall best performance when considering both F1-score and AUC (Table 7).

Table 3: Comparison of the precisions (%) of DeepSSInter and three other methods on the Heterodimer99 test set considering the top 1, 10, 25, 50, L/10, L/5, and L predicted contacts with the experimental sequences and structures as input. The data of the other methods are taken from the literature (Lin et al., 2023).

| Method | Top 1 | Top 10 | Top 25 | Top 50 | Top L/10 | Top L/5 | Top L |
|---|---|---|---|---|---|---|---|
| **DeepSSInter** | **59.6** | **50.0** | **48.8** | **45.7** | **50.8** | **48.8** | **41.9** |
| DeepInter | 45.5 | 46.1 | 44.7 | 43.7 | 44.9 | 44.4 | 40.0 |
| CDPred | 39.8 | 35.4 | 33.1 | 30.4 | 35.2 | 33.3 | 26.5 |
| GLINTER | 37.4 | 33.0 | 28.9 | 26.1 | 32.3 | 29.0 | 22.2 |

We can see from Table 3 that out of the other three methods, DeepInter is the best-performing. Compared with DeepInter, DeepSSInter performs significantly better, achieving a 1.9-14.1% improvement for top-k precisions. This improvement suggests that DeepSSInter's structure-aware features are able to better capture patterns that are relevant to interface contact prediction not only for homodimeric complexes, but also for heterodimeric complexes.

Compared with the other two methods (excluding DeepInter), DeepSSInter also achieves improvements of 14.6-19.8% and 17.0-22.2% compared to CDPred and GLINTER, respectively. Therefore, we can see that DeepSSInter is also able to improve the interface prediction accuracies for the challenging heterodimeric protein complexes compared to the other methods. Especially, the great improvement of DeepSSInter on the top-1 precision compared with other methods, will be of great benefit on the predictions of protein complex structures by the protein docking algorithms.

Similar to the homodimer evaluations, we also tested our model on AlphaFold2-predicted heterodimeric protein structures as input to investigate the robustness of our method. We used the same Heterodimer99 test set, but instead of using experimental sequences and structures as input, we used the full-length sequences and AlphaFold2-predicted structures. It is noted that only 95 complexes are tested here because AlphaFold2 failed on two complexes and also two other complexes contain non-standard amino acids. The precisions of the four models when inputting AlphaFold2-predicted structures are shown Table 4. It can be seen from the table that DeepSSInter achieves top-k precisions of 34.7%, 32.4%, 31.3%, 30.4%, 32.2%, 31.5%, and 27.3% for k values of 1, 10, 25, 50, L/10, L/5, and L respectively. In addition, DeepSSInter also achieves the best performance in terms of F1-score and AUC (Table 8). When comparing top L precisions of the four models, we can see that DeepSSInter, with top L precision of 27.3%, performs better than CDPred and Glinter, which obtain top L precisions of 22.8% and 19.3% respectively. Again, DeepSSInter outperforms three typical structure prediction methods including HDOCKlite, ESM-FOLD, and AlphaFold-Multimer w/o MSA in predicting inter-protein contacts of heterodimers (Table 4).

Interestingly, DeepSSInter performs worse than DeepInter with the top L precisions of 27.3% versus 29.4%. This indicates that DeepSSInter still lacks in some robustness compared to DeepInter, but is more robust than CDPred and GLINTER in general. The reason of the lack in performance on AlphaFold2-predicted structures compared to DeepInter may be the information loss in single-sequence protein language models. Especially for heterodimeric complexes for which the prediction of structure is already more difficult, passing the full-length sequence of the heterodimer complex into the protein models may lead to features that have some discrepancy, causing lower prediction precision. In such cases, paired MSA for DeepInter may better capture the coevolutionary features than single-sequence protein language models like ESM for DeepSSInter. However, it should also be noted that DeepSSInter is much faster and more scalable than DeepInter because DeepSSInter does not rely on the input of MSA. Therefore, DeepSSInter still owns an overall benefit compared with DeepInter by weighing their speed and accuracy.

Table 4: Comparison of the precisions (%) of DeepSSInter and three other methods on the Heterodimer99 test set considering the top 1, 10, 25, 50, L/10, L/5, and L predicted contacts with the full sequences and AlphaFold2-predicted structures as input. The data of the other methods are taken from the literature (Lin et al., 2023).

| Method | Top 1 | Top 10 | Top 25 | Top 50 | Top L/10 | Top L/5 | Top L |
|---|---|---|---|---|---|---|---|
| DeepSSInter | 34.7 | 32.4 | 31.3 | 30.4 | 32.2 | 31.5 | 27.3 |
| **DeepInter** | **42.1** | **37.4** | **35.5** | **33.8** | **36.4** | **35.3** | **29.4** |
| CDPred | 34.7 | 32.0 | 29.8 | 27.0 | 32.2 | 30.4 | 22.8 |
| GLINTER | 35.8 | 27.5 | 25.3 | 23.0 | 27.0 | 24.8 | 19.3 |
| HDOCKlite | 25.8 | 25.2 | 24.7 | 24.6 | 25.1 | 25.0 | 24.0 |
| ESMFold | 27.3 | 29.4 | 29.9 | 29.7 | 30.4 | 30.2 | 28.0 |
| AFM w/o MSA | 7.5 | 7.4 | 7.5 | 8.0 | 7.2 | 7.1 | 7.5 |

## 4.4 APPLICATION TO REALISTIC CASP-CAPRI COMPLEXES

To evaluate DeepSSInter in real applications, we also tested DeepSSInter on an additional test set of realistic CASP_CAPRI complexes (Lin et al., 2023). As shown in Appendix C, DeepSSInter also outperforms the other methods for top L predicted contacts (Tables 9 and 10), demonstrating the accuracy and robustness of DeepSSInter.

## 4.5 ABLATION STUDY

To investigate the effect of each component within the model architecture and verify their effectiveness, we conducted ablation studies on DeepSSInter. We trained five new models by removing the geometric transformer module (no_gt), ESM2 module (no_ESM), SaProt module (no_SaProt), both the geometric transformer and ESM2 modules (no_gt_ESM), and both the geometric transformer and SaProt modules (no_gt_SaProt), respectively. All of the five ablation models are trained with the same hyperparameters as the baseline model.

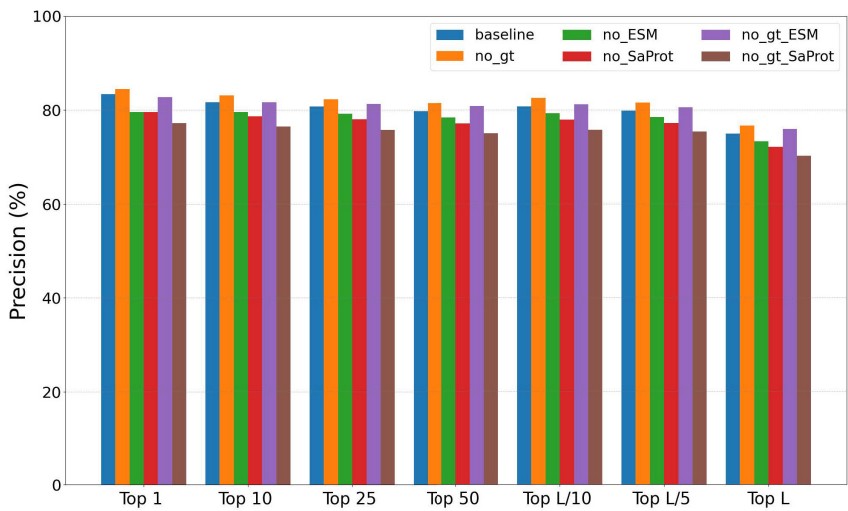

Figure 3: The performance for the ablation models versus the baseline model for several top numbers of predicted contacts on the Homodimer289 test set with experimental structures as input.

When testing the ablation models on the Homodimer289 test set, we can see from Figure 3 that no_gt slightly improves precisions, while all other ablation models (no_ESM, no_SaProt, no_gt_ESM, and no_gt_SaProt) have lower precisions than the baseline model. When testing the ablation models on the Heterodimer99 test set, we can see from Figure 4 that all ablation models (no_gt, no_ESM,

no_SaProt, no_gt_ESM, and no_gt_SaProt) have lower precisions than the baseline model. Similar trends can be observed in terms of F1-score and AUC of different ablation models (Tables 11 and 12), and also shown in their contact maps (Figure 5).

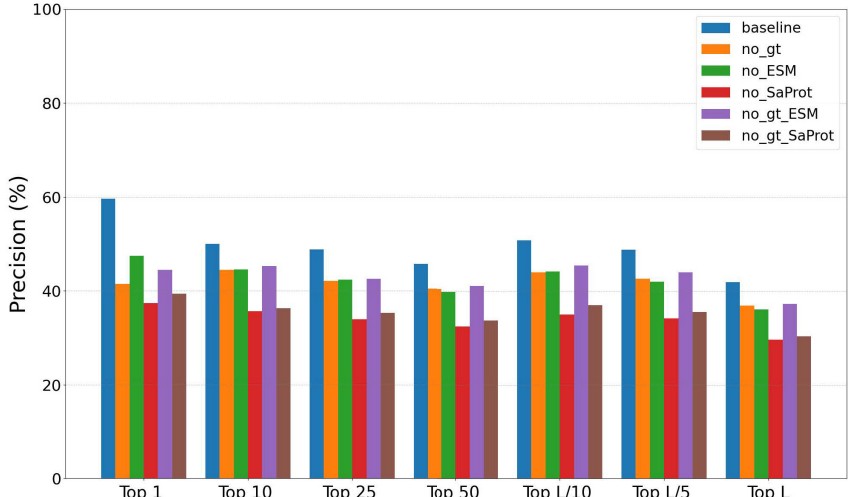

Figure 4: The performance for the ablation models versus the baseline model for several top numbers of predicted contacts on the Heterodimer99 test set with experimental structures as input.

The ablation experiments on the Homodimer289 and Heterodimer99 test sets demonstrate the importance of integrating ESM2 and SaProt protein language models. Overall, SaProt is the most impacting factor, following by GT (geometric transformer) and ESM. This can be understood because SaProt is built on ESM and may implicitly include the features of ESM (Su et al., 2023). The phenomenon that the geometric transformer slightly decreases the performance for homodimers is possibly due to the interplay between the graph and distance representations of monomer structures. Compared with distance representation, graph representation is normally less precise but more robust against structural errors. As such, geometric transformer may not help the model for high-accuracy homodimer cases that are determined by co-evolutions, but would play a significant role for medium or low-accurate heterodimer cases that are largely determined by structural features. How to balance the structure representations from graph transformer and SaProt protein language model remains an important topic in the future development of DeepSSInter.

## 5  CONCLUSION

We have proposed a sequence and structure-aware protein language-based deep learning model to effectively predict the interface contacts for protein-protein interactions, named DeepSSInter. Compared with state-of-the-other methods such as DeepInter, DeepHomo2.0, GLINTER, CDPred, and DeepHomo, our DeepSSInter model achieves the best performance for all precision metrics on diverse test sets of homodimeric and heterodimeric protein complexes, respectively, when utilizing experimental protein structures as input. On average, our DeepSSInter method achieves a top L/5 prediction of 79.8% on the homodimeric complexes, compared with 77.1% for DeepInter, 67.9% for CDPred, 67.2% for DeepHome2.0, 57.4% for GLINTER, and 52.0% for DeepHomo, respectively. On the heterodimeric complexes, DeepSSInter obtains a top L/5 precision of 48.8%, which is significantly higher than 44.4% for DeepInter, 33.3% for CDPred, and 29.0% for GLINTER, respectively. In addition, our model also performs well on AlphaFold2-predicted structures, showing its robustness on predicted structures. Despite DeepSSInter's high precision and robustness, there still exist some limitations in the model, such as the difficulty in predicting the inter-protein contacts of heterodimers with AlphaFold2-predicted structure. However, our model shows the effectiveness of using protein language models and structure-aware features in improving the accuracy of predicting the interface contacts of protein complexes.

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

:

# A   COMPARISON OF DEEPSSINTER WITH OTHER METHODS IN TERMS OF F1-SCORE AND AUC ON THE HOMODIMER289 TEST SET.

Table 5: Comparison of DeepSSInter with other methods in terms of F1-score and AUC (area under the ROC) of contact prediction on the Homodimer289 test set with the *experimental* sequences and structures as input.

| Method | F1-score | AUC |
| --- | --- | --- |
| DeepSSInter | **0.5993** | **0.9718** |
| DeepInter | 0.4618 | 0.9716 |
| CDPred | 0.3574 | 0.9467 |
| GLINTER | 0.2762 | 0.8927 |
| DeepHomo2 | 0.2440 | 0.9413 |
| DeepHomo | 0.0669 | 0.9185 |

Table 6: Comparison of DeepSSInter with other methods in terms of F1-score and AUC (area under the ROC) of contact prediction on the Homodimer289 test set with the *AlphaFold2-predicted* structures as input.

| Method | F1-score | AUC |
| --- | --- | --- |
| DeepSSInter | **0.4879** | 0.9268 |
| DeepInter | 0.3822 | **0.9294** |
| CDPred | 0.3728 | 0.9187 |
| GLINTER | 0.2532 | 0.8781 |
| DeepHomo2 | 0.1963 | 0.9030 |
| DeepHomo | 0.0470 | 0.8886 |

# B COMPARISON OF DEEPSSINTER WITH OTHER METHODS IN TERMS OF F1-SCORE AND AUC ON THE HETERODIMER99 TEST SET.

Table 7: Comparison of DeepSSInter with other methods in terms of F1-score and AUC (area under the ROC) of contact prediction on the Heterodimer99 test set with the *experimental* sequences and structures as input.

| Method | F1-score | AUC |
|---|---|---|
| DeepSSInter | **0.2259** | 0.8914 |
| DeepInter | 0.1832 | **0.8960** |
| CDPred | 0.0691 | 0.8267 |
| GLINTER | 0.0834 | 0.8148 |

Table 8: Comparison of DeepSSInter with other methods in terms of F1-score and AUC (area under the ROC) of contact prediction on the Heterodimer99 test set with the *AlphaFold2-predicted* structures as input.

| Method | F1-score | AUC |
|---|---|---|
| DeepSSInter | **0.1479** | **0.8355** |
| DeepInter | 0.1229 | 0.8035 |
| CDPred | 0.0540 | 0.7715 |
| GLINTER | 0.0857 | 0.8071 |

# C  COMPARISON OF DEEPSSINTER WITH OTHER METHODS ON CASP-CAPRI COMPLEXES

Table 9: Comparison of the precisions (%) of DeepSSInter and other methods on the CASP-CAPRI test set of 27 complexes considering the top 1, 10, 25, 50, L/10, L/5, and L predicted contacts with the *experimental* sequences and structures as input. The data of the other methods are taken from the literature (Lin et al., 2023).

| Method | Top 1 | Top 10 | Top 25 | Top 50 | Top L/10 | Top L/5 | Top L |
|---|---|---|---|---|---|---|---|
| DeepSSInter | 63.0 | 65.9 | 64.9 | 64.7 | 65.3 | 64.3 | **64.3** |
| DeepInter | **74.1** | **71.1** | **71.4** | **69.6** | **71.0** | **69.3** | 61.7 |
| CDPred | 66.7 | 67.8 | 64.1 | 63.0 | 65.1 | 62.9 | 51.6 |
| GLINTER | 70.4 | 64.8 | 63.0 | 60.4 | 62.4 | 59.0 | 45.6 |
| DeepHomo2 | 70.4 | 64.8 | 63.0 | 60.4 | 62.4 | 59.0 | 45.6 |
| DeepHomo | 55.6 | 50.7 | 46.4 | 43.6 | 44.5 | 43.0 | 30.7 |

Table 10: Comparison of the precisions (%) of DeepSSInter and other methods on the CASP-CAPRI test set of 27 complexes considering the top 1, 10, 25, 50, L/10, L/5, and L predicted contacts with the *AlphaFold2-predicted* structures as input. The data of the other methods are taken from the literature (Lin et al., 2023).

| Method | Top 1 | Top 10 | Top 25 | Top 50 | Top L/10 | Top L/5 | Top L |
|---|---|---|---|---|---|---|---|
| DeepSSInter | 63.0 | **66.3** | **65.3** | **65.0** | **65.9** | **65.3** | **65.3** |
| DeepInter | 66.7 | 63.7 | 63.3 | 62.7 | 64.5 | 63.4 | 55.9 |
| CDPred | – | 65.7 | – | – | 63.1 | 60.9 | 49.0 |
| GLINTER | 63.0 | 62.2 | 58.7 | 54.1 | 58.5 | 53.4 | 35.6 |
| DeepHomo2 | **70.4** | 63.7 | 59.1 | 56.4 | 58.1 | 55.6 | 42.4 |
| DeepHomo | 59.3 | 53.7 | 47.4 | 44.6 | 46.1 | 43.8 | 29.4 |

# D Ablation experiments in terms of F1-score and AUC.

Table 11: The performance for the ablation models versus the baseline model in terms of F1-score and AUC (area under the ROC) of contact prediction on the *Homodimer289* test set with the experimental sequences and structures as input.

| Method | F1-score | AUC |
|---|---|---|
| baseline | 0.5993 | **0.9718** |
| no_gt | **0.6078** | 0.9707 |
| no_esm | 0.5935 | 0.9706 |
| no_saport | 0.5423 | 0.9566 |
| no_gt_esm | 0.5950 | 0.9725 |
| no_gt_saprot | 0.5558 | 0.9602 |

Table 12: The performance for the ablation models versus the baseline model in terms of F1-score and AUC (area under the ROC) of contact prediction on the *Heterodimer99* test set with the experimental sequences and structures as input.

| Method | F1-score | AUC |
|---|---|---|
| baseline | **0.2259** | **0.8914** |
| no_gt | 0.1824 | 0.8726 |
| no_esm | 0.1923 | 0.8868 |
| no_saport | 0.1413 | 0.8240 |
| no_gt_esm | 0.1783 | 0.8855 |
| no_gt_saprot | 0.1376 | 0.8109 |

# E CONTACT MAPS PREDICTED BY THE BASELINE AND ABLATION MODELS OF DEEPSSINTER.

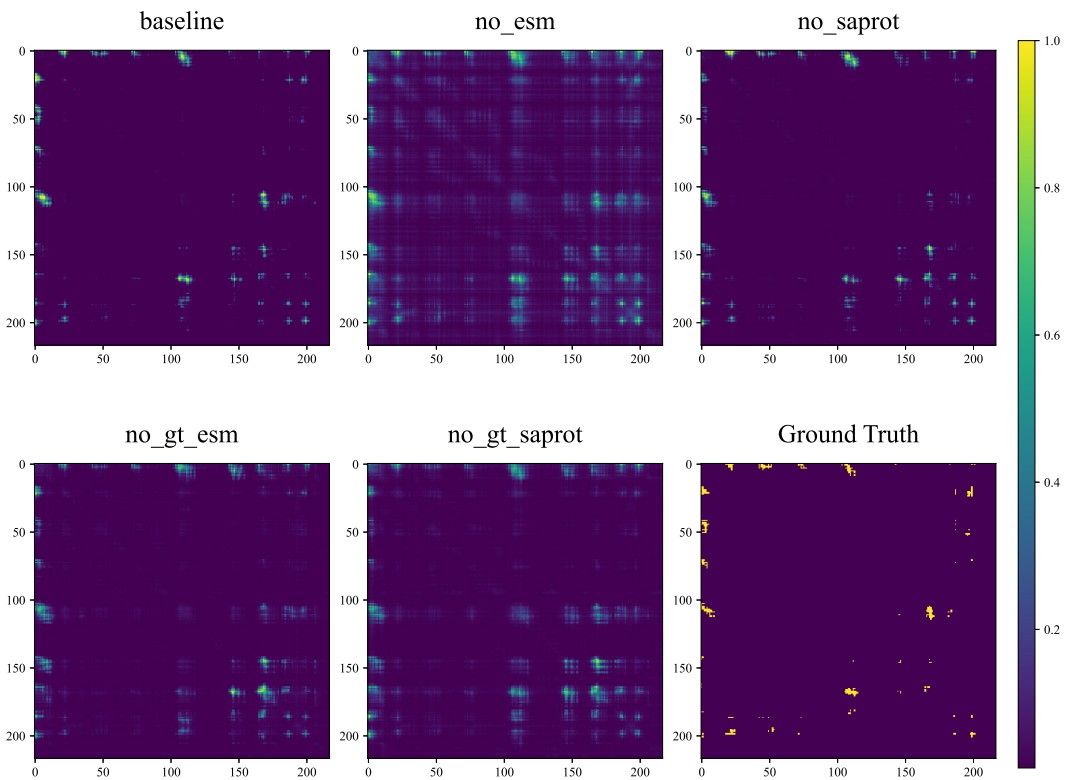

Figure 5: Contact maps predicted by the baseline and ablation models of DeepSSInter on an example complex (PDB code: 2WAG). The baseline model gives a topL precision of 88.5%, compared with 56.7% for no_gt, 73.3% for no_esm, 40.6% for no_saprot, 69.6% for no_gt_esm , and 39.2% for no_gt_saprot.

## F  HOW TO USE DEEPSSINTER AT THE PREDICTION TIME.

At the prediction time, only one (for a homodimer case) or two (for a heterodimer case) monomer structures are needed as input for DeepSSInter. By default, DeepSSInter does not crop the protein structure and use the full sequence to predict residue-residue contacts during the inference. For the best performance of DeepSSInter, it is not recommended that users crop the protein structures at the prediction time. If a very long protein (e.g. $> 5000aa$) causes an overflow of GPU memory at the prediction time, users may run DeepSSInter on CPU. Nevertheless, if users really need to crop a protein due to memory and/or speed reason at the prediction time, they may use a sliding window strategy. In such cases, it is recommended to use a window size as long as allowed by users' computer or choose a window size that have an opportunity to cover more interface residues during sliding. As shown in Figure 6, if a cropped structure can cover more interface residues, the top predicted contact tends to have a higher contact probability, which may be used to guide the choice of a window size. Moreover, if a cropped structure can cover more interface residues, DeepSSInter also tends to have a higher precision in contact prediction (Figure 7). There results demonstrate the feasibility of using a sliding window strategy for very long proteins at the prediction time.

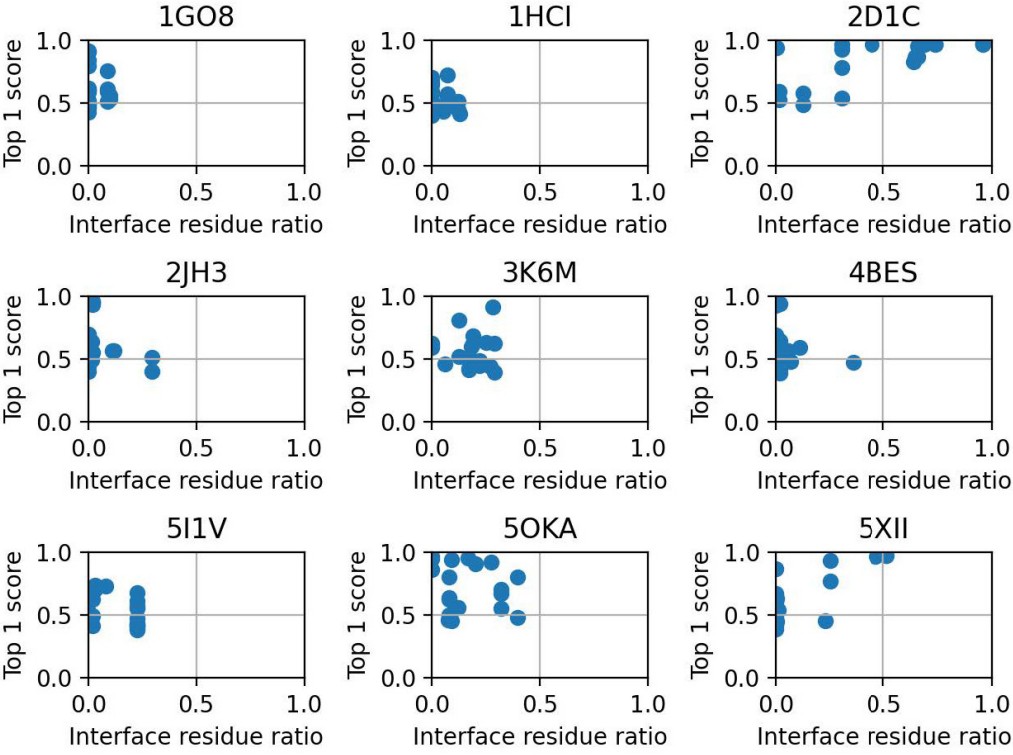

Figure 6: The top 1 score (i.e. the contact probability of the top predicted contact by DeepSS-Inter) versus the interface residue ratio of a cropped structure on nine homodimer examples. For demonstration purpose, the widow size is here set to 200 aa. For each case, 20 protein structures are cropped from the full-length monomer by evenly sliding the window.

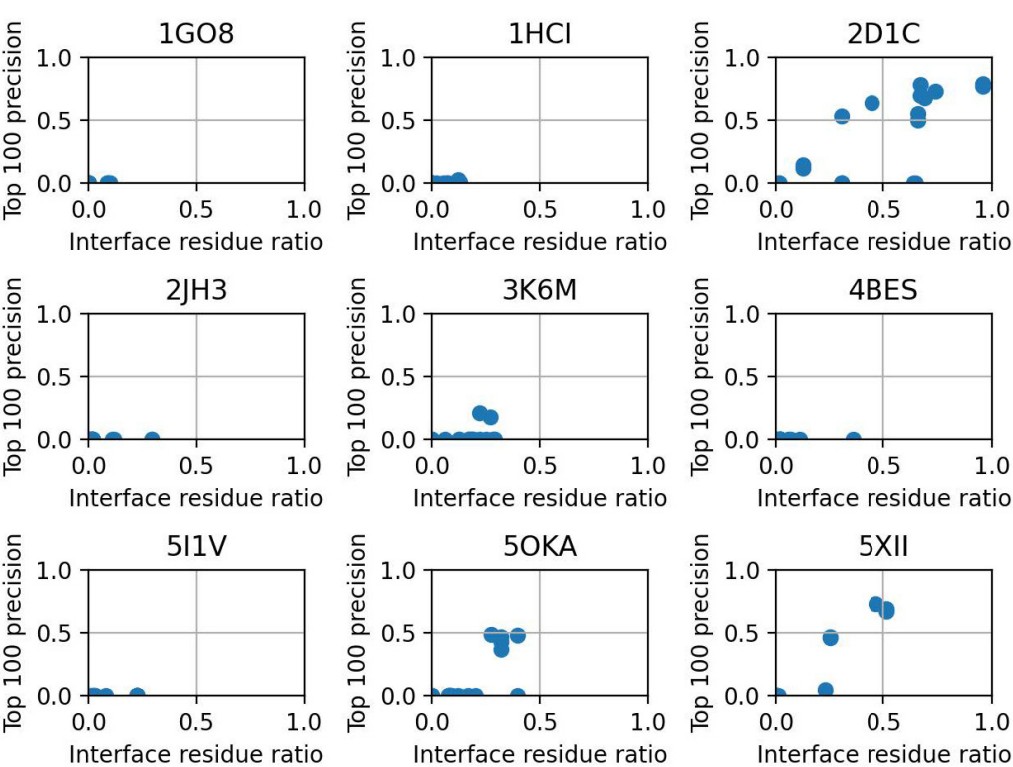

Figure 7: The top 100 precision of contact prediction by DeepSSInter versus the interface residue ratio of a cropped structure on nine homodimer examples. For demonstration purpose, the widow size is here set to 200 aa. For each case, 20 protein structures are cropped from the full-length monomer by evenly sliding the window.

