# OpenReview forum: "Prediction of Protein-protein Contacts with Structure-aware Single-sequence Protein Language Models"
_ICLR.cc/2025/Conference — Submitted to ICLR 2025_

### Official Review · Reviewer_aiEL · 2024-10-28

**Soundness:** 3
**Presentation:** 3
**Contribution:** 3
**Rating:** 6
**Confidence:** 3

**Summary:**

In this paper, the authors present a state-of-the-art model for protein-protein contact map predictions, without the use of MSA, which indeed are hard to come by in the case of heteromers and represent a big improvement in tackling protein-protein contact map problem.
To do so they essentially built a model that processed each binding  partner separately as well as the complex formed by the partners, and  they did it using different types of information:
- An EGNN is used to discover useful amino acid representations of both partners as separate entities, from structure information.
- A RBF is used to enrich the description of distances between residue at different scales, leading to the building of an intra-distance matrix for both partners as separate entities.
- ESM embeddings are used to derive useful amino acid representations of both partners as separate entities, from evolutionary information.
- ESM attention is used as a counterpart for the distance-based matrix from the RBF transformation.
- The same that was done using ESM is now done using SaProt for retrieval of amino acid information more focused on the local structural environment.
- Finally, both ESM-derived features and SaProt-derived features are also produced for the partner pair. For ESM-related features, it comes with the addition of a linker, and for SaProt, a sequence of unknown tokens as foldseek has no complex structures to work with.
All of those part are then run through a ResNet Inception module followed by a triangle aware module, to produce a protein-protein contact map.

**Strengths:**

Apart from the fact that authors present a model that reaches state-of-the-art performance in most conditions and that it doesn't need MSA (and that's not a small achievement), the paper's strength lays in how it breaks down its conclusions: homodimers, heterodimers, real pdbs, alphafold-generated pdbs, and ablation study. The study by pdb type is super interesting, and seems to show that this model is still useful in the alphafold pdb case but probably more in the homodimeric case, and in the heterodimer case one can still work with this model MSA-based counterpart model.

**Weaknesses:**

This paper has 3 main caveats which I believe are not deal-breakers but for which at least 2 would (and could) benefit for improvement in how it is presented in the paper.
The first one is that it is a very massive pipeline and understanding what really matters seems hard.
The second one is the lack of a clear description of how to use the model at prediction time.
The last point is about the overall presentation of results and data, mainly incorporating more evaluation metrics and a bit deeper description of the data involved.

**Questions:**

Indeed, it is quite hard to make sense of the ablation study (not the author's fault, the results are what they are). For example for the homodimer case: no_gt_esm as good or even better than the full model? Reverse conclusion for heterodimers? I left the reading with the feeling: it works great but is all of that really necessary? Of course, having a (performant !) model straying away from MSA is very welcomed and necessary, but what matters in the model is not obvious. A way maybe to add to this ablation study would be to have a comparison (just plotting side by side with some metrics for example: F1 score , area under the ROC, difference between contact map, where is the difference on the structure ) for a few complexes for different models.

The second one is the lack of a clear description of how to use the model at prediction time. By that I mean the following. Sequences have to be truncated when they are too big, for obvious memory and time complexity reasons, but those truncations, at least given how they are explained, rely on some ground truth of a sequence window that has maximum PPI. At prediction time if we have a pair of big proteins and no idea how to choose this window how do we do? Moreover, how can we assess the confidence of those pairings in the contact map? Could we have then a study where for long protein sliding or random windows are chosen and distribution of predicted contact residue numbers are plotted: in that case is the model most of the time predicting the maximum number of contact at the right ground truth window? How often does it not? Is this a strategy at prediction time, and can we use this distribution to have a confidence about the window chosen?

The last point is about the overall presentation of results and data:
- Results: Could we have more metrics (F1 score, area under the ROC) than the precision which would give more information about the different types of errors? Maybe a few confusion matrices for some cases where the model works fine and for when the model works worse. Also, a few of those contact maps predicted as well as how that translates on the structures. I find it really hard to draw a good picture from a paper when only one metric (a statistical one in this case) is used for the whole discussion.
- Data: Some data analysis of the training data. For example, I have no idea what is the distribution of number of contact interactions given some protein length. Mainly as the sequence has to be cropped around part of the sequence with maximum contact with the other protein partner, I would like to have an idea of what that maximum represents:  are we still in the super imbalanced class regime, how easy it is to reach 80% precision in that case by just putting every pairs of residue  as contact residue and so on. Is this cropping strategy leading to a very different distribution of contact fraction than the non-cropped one? How sensitive is it to this choice of 320 residues and so on...

Side questions:
Would a more stringent but identical preprocessing step for both pdb types (real vs alphafold) be a possible way to reduce the drop in performance?
For the drop of performance between homodimers and heterodimers, did the author try to bump up the loss weight for heterodimers?

---

> ### Author Response · Authors · 2024-11-23
> **Responses to Reviewer aiEL**
>
> We thank the reviewer for reviewing our manuscript and giving the valuable questions. We have carefully addressed the weaknesses/questions. Following the reviewer’s suggestion, we have added more evaluation metrics including F1-score and AUC as well as the contact maps for a deeper description of the data involved (**please see Appendix A, B, D, and E for the results**). Indeed, the results by more metrics demonstrate many more advantages of our DeepSSInter model over other methods.
>
> Our manuscript has been updated accordingly. We really appreciate the reviewer’s valuable time and comments, and hope the reviewer can reevaluation our study considering our responses below.
>
> _**Weaknesses:**_
>
> _This paper has 3 main caveats which I believe are not deal-breakers but for which at least 2 would (and could) benefit for improvement in how it is presented in the paper._
>
> _**Weakness 1:** The first one is that it is a very massive pipeline and understanding what really matters seems hard._
>
> **Response:** We thank the reviewer for the comment. The reason is that the goal of this study is to propose an efficient deep learning-based integrative model, and then apply it to solve the inter-protein residue-residue contact problem in the biology field. Apparently, a simple pipeline may not work well due to the complexity of the problem. In fact, our model needs to leverage the embedding features of different protein language model. In addition, we also utilize the geometric graph transformer to process the protein structure for extracting the structure representation to improve the accuracy of our network. The ablation study also demonstrates the importance of each of the modules in our pipeline
>
> _**Weakness 2:** The second one is the lack of a clear description of how to use the model at prediction time._
>
> **Response:** We thank the reviewer for the comment. We have added a description of how to use the model at prediction time in page 4 of our manuscript as “During the prediction time, only one (for a homodimer case) or two (for a heterodimer case) monomer structures are needed as input for the model. The output is a contact map consisting of the pairwise probabilities between the amino acids of two proteins.”
>
> _**Weakness 3:** The last point is about the overall presentation of results and data, mainly incorporating more evaluation metrics and a bit deeper description of the data involved._
>
> **Response:** We thank the reviewer for the comment and suggestions. We have added more evaluation metrics including F1-score and AUC for a deeper description of the data involved.
>
> Pease see the main text and **Appendix A, B, D, and E** in our manuscript for detailed revisions.
>
>
> _**Questions:**_
>
> _**Questions 1:** Indeed, it is quite hard to make sense of the ablation study (not the author's fault, the results are what they are). For example for the homodimer case: no_gt_esm as good or even better than the full model? Reverse conclusion for heterodimers? I left the reading with the feeling: it works great but is all of that really necessary? Of course, having a (performant !) model straying away from MSA is very welcomed and necessary, but what matters in the model is not obvious. A way maybe to add to this ablation study would be to have a comparison (just plotting side by side with some metrics for example: F1 score , area under the ROC, difference between contact map, where is the difference on the structure ) for a few complexes for different models._
>
> **Response:** We thank the reviewer for the valuable comment and suggestions. We have added more evaluation metrics including F1-score and AUC as well as the contact maps of a dimer example in the ablation study (**Pease see the Appendix D and E in the manuscript**), and discussed the impacting factors as follows.
>
> “Overall, SaProt is the most-impacting factor, following by GT (geometric transformer) and ESM. This can be understood because SaProt is built on ESM and may implicitly include the features of ESM. The phenomenon that the geometric transformer slightly decreases prediction performance for homodimers is possibly due to the interplay between the graph and distance representations of monomer structures. Compared with distance representation, graph representation is normally less precise but more robust against structural errors. As such, geometric transformer may not help the model for high-accurate homodimer cases that are determined by co-evolutions, but would play a significant role for medium or low-accurate heterodimer cases that are largely determined by structural features. How to balance the structure representations from graph transformer and SaProt protein language model remains an important topic in the future development of DeepSSInter.”

---

> ### Author Response · Authors · 2024-11-24
> **Responses to Reviewer aiEL (continue)**
>
> _**Question 2:** The second one is the lack of a clear description of how to use the model at prediction time. By that I mean the following. Sequences have to be truncated when they are too big, for obvious memory and time complexity reasons, but those truncations, at least given how they are explained, rely on some ground truth of a sequence window that has maximum PPI. At prediction time if we have a pair of big proteins and no idea how to choose this window how do we do? Moreover, how can we assess the confidence of those pairings in the contact map? Could we have then a study where for long protein sliding or random windows are chosen and distribution of predicted contact residue numbers are plotted: in that case is the model most of the time predicting the maximum number of contact at the right ground truth window? How often does it not? Is this a strategy at prediction time, and can we use this distribution to have a confidence about the window chosen?_
>
> **Response:** We thank the reviewer for the valuable questions. We have added a description of how to use the model at prediction time as “During the prediction time, only one (for a homodimer case) or two (for a heterodimer case) monomer structures are needed as input for the model. The output is a contact map consisting of the pairwise probabilities between the amino acids of two proteins.” (**page 4 in the manuscript**)
>
> In addition, it is a great suggestion and idea to find a suitable slice of the sequence in the inference for large sequence lengths. In the training stage, we choose the window with the largest inter-protein contacts for containing the interacting interface and balance the positive and negative of labels. In the inference stage, it may have some situations where the required GPU memory is insufficient. Therefore, it needs to choose a suitable window. For example, in DeepHomo2, the predicted probability of pairwise residue has a larger value, it would have a high confidence to be a true interface contact. Therefore, we can use the sliding window to find the maximum predicted inter-protein contact probability.
>
> _**Results:** Could we have more metrics (F1 score, area under the ROC) than the precision which would give more information about the different types of errors? Maybe a few confusion matrices for some cases where the model works fine and for when the model works worse. Also, a few of those contact maps predicted as well as how that translates on the structures. I find it really hard to draw a good picture from a paper when only one metric (a statistical one in this case) is used for the whole discussion._
>
> **Response:** We thank the reviewer for the valuable suggestions. We have added the results of more metrics including F1-Score and AUC of different methods (Please see Appendix A, B, and D). Indeed, the results with these metrics demonstrate many more advantages of our DeepSSInter model over other methods.
>
> _**Data:** Some data analysis of the training data. For example, I have no idea what is the distribution of number of contact interactions given some protein length. Mainly as the sequence has to be cropped around part of the sequence with maximum contact with the other protein partner, I would like to have an idea of what that maximum represents: are we still in the super imbalanced class regime, how easy it is to reach 80% precision in that case by just putting every pairs of residue as contact residue and so on. Is this cropping strategy leading to a very different distribution of contact fraction than the non-cropped one? How sensitive is it to this choice of 320 residues and so on..._
>
> **Response:** In the vast majority of cases, the ratio of positive and negative labels in the inter-protein contact map is less than 1%. Therefore, for the balance of positive and negative labels, and the maximum interacting interface, the choice of maximum contacts would benefit for training. Even if we crop the sequence by this strategy, the ratio of positive and negative label is still large. In addition, within the limits of GPU memory requirements, maximizing the upper limit of crop length will help the model leverage more sequence information.
>
> _**Side questions:** Would a more stringent but identical preprocessing step for both pdb types (real vs alphafold) be a possible way to reduce the drop in performance? For the drop of performance between homodimers and heterodimers, did the author try to bump up the loss weight for heterodimers?_
>
> **Response:** We thank the reviewer for the valuable questions. In the training stage of DeepInter, it separately trains two models for homodimer and heterodimer. In the current method, we merge the two types of complexes for training, and achieve better and similar performance with DeepInter. Therefore, choosing a different loss weight may have a little effect. However, it might be possible to find a more suitable loss weight to slightly improve the performance.

---

> > ### Comment · Reviewer_aiEL · 2024-11-25
> >
> > Dear Authors,
> > Thanks a lot for you answers.
> > Most of my questions have been answered at least to some degree and I really appreciate the authors hard work there. I see that the sliding window strategy is one that you would explore at prediction time. I still feel that this is worth mentioning in the main text (let me know if I have missed it), and worth showing an output of such strategy in appendix : like a distribution of predicted contact over this sliding window, just to show if it is easy to define some group of windows in a test case and how it translates when comparing to ground truth.
> > I realized that I might seem a bit laser focus on this point: but I believe the strength of this paper is in its usefulness (less in what we learn from it), so having a "real" test case scenario would strengthen it.
> > Looking forward to your answer.

---

> > > ### Author Response · Authors · 2024-11-27
> > > **Response to Reviewer aiEL**
> > >
> > > >Dear Authors, Thanks a lot for you answers. Most of my questions have been answered at least to some degree and I really appreciate the authors hard work there. I see that the sliding window strategy is one that you would explore at prediction time. I still feel that this is worth mentioning in the main text (let me know if I have missed it), and worth showing an output of such strategy in appendix : like a distribution of predicted contact over this sliding window, just to show if it is easy to define some group of windows in a test case and how it translates when comparing to ground truth. I realized that I might seem a bit laser focus on this point: but I believe the strength of this paper is in its usefulness (less in what we learn from it), so having a "real" test case scenario would strengthen it. Looking forward to your answer.
> > >
> > > **Response:** We thank the reviewer for reviewing our revised manuscript and give the positive comments. We are sorry for missing the important point about the sliding window strategy. Following the reviewer’s suggestion, we have added an appendix section (**Appendix F**) to describe how to use DeepSSInter at the prediction time, and demonstrated the feasibility of using a sliding window strategy for very long proteins at the prediction time. We also add to two figures to show the results of cropped structures from the full-length monomer by evenly sliding a window on nine homodimer test cases (**please see Figures 6 and 7**). The manuscript has been updated accordingly.
> > >
> > > In addition, we mentioned the sliding window strategy in the manuscript as
> > >
> > > “By default, DeepSSInter does crop the monomer structure during the inference. However, for a very long protein, users may use a sliding window strategy (Appendix F).” (**please see page 4 of the manuscript**).
> > >
> > > We also gave a detailed description about “How to use DeepSSInter at the prediction time” in the **Appendix F** of the manuscript.
> > >
> > > We hope we have addressed the reviewer’s concerns. If we missed something or mis-understanding the reviewer’s point, please don’t hesitate to let us know. We will try to address the reviewer’s concerns/questions as possible as we can.
> > >
> > > Again, we thank the reviewer for reviewing our study and appreciate the reviewer’s valuable time.

---

> > > > ### Comment · Reviewer_aiEL · 2024-11-27
> > > >
> > > > Dear authors,
> > > > Thanks a lot for your answer. It sets nicely the chain length range for which the model can be directly applied and provides some insight about the model capacity outside of the direct use case scenario. The trend on those plots are not as obvious as I was expecting, but as I said it gives future users an understanding of what they can expect if they are outside of this max length range and I believe that's a nice addition to the paper.

---

### Official Review · Reviewer_xbjr · 2024-10-28

**Soundness:** 3
**Presentation:** 2
**Contribution:** 3
**Rating:** 8
**Confidence:** 4

**Summary:**

The paper presents DeepSSInter, a novel transformer-based deep learning model designed to predict protein-protein interface contacts using structure-aware single-sequence protein language models. This model's efficiency and predictive power mark a significant advancement in protein interaction prediction, particularly beneficial for cases where MSA is unavailable or inefficient.

**Strengths:**

Originality: The model creatively combines advanced components like graph-based features, ResNet-Inception, and a triangle-aware module, innovatively moving beyond MSA. These elements effectively address challenges in both computational complexity and prediction accuracy, especially for heterodimers, which are historically difficult for MSA-based models.

Quality: The authors validate DeepSSInter rigorously through extensive experiments on homodimeric and heterodimeric datasets, consistently outperforming five state-of-the-art methods in precision metrics. Ablation studies and evaluations with AlphaFold2-predicted structures further strengthen the model’s demonstrated robustness and confirm the necessity of each architectural component.

Clarity: The paper is well-organized, with clear descriptions of model components, experimental setup, and visual aids like architecture figures. Though some technical terms could use more explanation, the methodology and results are easy to follow.

Significance: By eliminating the need for MSA, DeepSSInter is both faster and more accurate, marking a significant advancement for applications in structural biology and drug discovery. This approach paves the way for single-sequence modeling in protein interaction research, making it impactful for high-throughput and resource-efficient studies.

**Weaknesses:**

1. Model Limitations with AlphaFold2-Predicted Structures
DeepSSInter's performance declines with AlphaFold2-predicted structures, especially for heterodimeric complexes, where it underperforms compared to DeepInter. This discrepancy suggests that single-sequence protein language models may lack the coevolutionary information MSA provides, which is crucial for some complex predictions. The paper would benefit from deeper analysis on this limitation, perhaps through: Alternative or augmented data inputs: Exploring ways to integrate low-dimensional coevolutionary features from paired MSAs, even minimally, could help bridge the gap between structure-aware single-sequence models and MSA-based predictions. Fine-tuning: Considering an additional fine-tuning phase on AlphaFold2-generated complexes could improve robustness to predicted structures, enhancing real-world applicability.
2. Limited Interpretability of Model Outputs
Although the paper shows improved prediction precision, the interpretability of the model outputs remains unaddressed. For practical applications in structural biology, understanding why certain contacts are predicted is often as important as the predictions themselves. Including feature attribution methods, like attention weight visualizations or feature importance analyses on the ESM2 and SaProt modules, could provide insights into how the model makes predictions, aiding users in the interpretation of contact predictions.
3. Broader Benchmarking and Testing
The paper focuses on precision metrics in homodimeric and heterodimeric complexes, but testing could be broadened to include more diverse protein types, such as oligomeric or multimeric complexes. These more complex structures are increasingly common in real-world biological contexts. Expanding benchmarks to these protein forms could provide a fuller view of DeepSSInter’s performance and generalizability.

**Questions:**

1.Could you provide more interpretative insights into which features or attention patterns are most important for your model's predictions?
2.Given the reliance on specific protein language models (ESM2 and SaProt), have you evaluated the model’s adaptability to other
3.Have you considered evaluating DeepSSInter on other types of protein complexes, such as oligomeric or multimeric structures? If so, what were the outcomes?
4.Could you elaborate on the choice of using both ResNet-Inception and triangle-aware modules? How do these components interact, and do they provide overlapping or distinct contributions to model performance?
5.Given the performance decline with AlphaFold2-predicted structures, have you considered fine-tuning the model on such predicted structures? If so, what were the results?

---

> ### Author Response · Authors · 2024-11-23
> **Responses to Reviewer xbjr**
>
> We thank the reviewer for reviewing our manuscript and giving the valuable comments. We have carefully addressed the weaknesses/questions. Moreover, following the reviewer’s suggestion, we have evaluated our model on an additional test set of more realistic complexes from CASP-CAPRI experiments. The results again demonstrate the better performance of our DeepSSInter model than other methods.
>
> Our manuscript has been updated accordingly. We really appreciate the reviewer’s valuable time and comments, and hope the reviewer can reevaluate our study by considering our responses below.
>
> _**Weaknesses:**_
>
> _**Weakness 1:** Model Limitations with AlphaFold2-Predicted Structures DeepSSInter's performance declines with AlphaFold2-predicted structures, especially for heterodimeric complexes, where it underperforms compared to DeepInter. This discrepancy suggests that single-sequence protein language models may lack the coevolutionary information MSA provides, which is crucial for some complex predictions. The paper would benefit from deeper analysis on this limitation, perhaps through: Alternative or augmented data inputs: Exploring ways to integrate low-dimensional coevolutionary features from paired MSAs, even minimally, could help bridge the gap between structure-aware single-sequence models and MSA-based predictions. Fine-tuning: Considering an additional fine-tuning phase on AlphaFold2-generated complexes could improve robustness to predicted structures, enhancing real-world applicability._
>
> **Response:** We thank the reviewer for the valuable comments. In the current method, we utilize the hidden representations of ESM2 and SaProt protein language models to improve the performance on the contact prediction of protein complexes. Since the both PLMs are pre-trained on the single sequences, they would better to extract the coevolutionary information from the MSA of monomer proteins. However, it can also link the different sequences to extract the coevolutionary information of inter-proteins. Specifically, for the homodimers, the receptor and ligand have the same sequences, it would be easily to extract the inter-protein coevolutionary than heterodimers.
> We agree that single-sequence PLM cannot capture sufficient accurate coevolutionary information, which can indeed be extracted through MSA. However, the primary goal of current methods is to address the situation with limited sequences in MSA.
>
> In addition, it is a nice suggestion to fine-turn the model with AlphaFold-predicted structures to improve the robustness and enhance real-world applicability. Unfortunately, we may not be able to provide the fine-tuning results of the fine-tune model within such a short time frame because it is extremely time-consuming to predicted monomer structures with AlphaFold2 in the training set. Nevertheless, we have seen similar fine-turn or training strategies in other studies (Ref. SaProt), and their experiments have demonstrated that training using AF2-predicted structures does not yield significant improvements.
>
> _**Weakness 2:** Limited Interpretability of Model Outputs Although the paper shows improved prediction precision, the interpretability of the model outputs remains unaddressed. For practical applications in structural biology, understanding why certain contacts are predicted is often as important as the predictions themselves. Including feature attribution methods, like attention weight visualizations or feature importance analyses on the ESM2 and SaProt modules, could provide insights into how the model makes predictions, aiding users in the interpretation of contact predictions._
>
> **Response:** We thank reviewer for the valuable comments and suggestions. Unfortunately, due to the limited time, we may not be able to conduct an extensive investigation. Instead, we have tried to address the interpretability of the model outputs through the ablation experiments. (**Please see page 10 of the manuscript**)
>
> _**Weakness 3:** Broader Benchmarking and Testing The paper focuses on precision metrics in homodimeric and heterodimeric complexes, but testing could be broadened to include more diverse protein types, such as oligomeric or multimeric complexes. These more complex structures are increasingly common in real-world biological contexts. Expanding benchmarks to these protein forms could provide a fuller view of DeepSSInter’s performance and generalizability._
>
> **Response:** We thank reviewer for the valuable suggestion. we have evaluated our model on an additional test set of more realistic complexes from CASP-CAPRI experiments.
>
> Please see Tables 9 and 10 for the results on the CASP-CAPRI test set. As shown in Tables 9 and 10 in Appendix C, the results again demonstrate the overall better performance of our DeepSSInter model than other methods. The two tables are also listed as follows.

---

> ### Author Response · Authors · 2024-11-24
> **Responses to Reviewer xbjr (continue)**
>
> _**Questions:**_
>
> _**Question 1:** Could you provide more interpretative insights into which features or attention patterns are most important for your model's predictions?_
>
> **Response:** We thank reviewer for the valuable comments and suggestions. Unfortunately, due to the limited time, we may not be able to conduct an extensive investigation on these issues. Instead, we have tried to get more interpretative insights through the ablation experiments and discussed them in page 10 of our manuscript.
>
> As addition information, ESMPair[1] and other works[2] have introduced the interpretation of the attention weighs with the contact map.
>
> [1] https://doi.org/10.1093/bib/bbad221
>
> [2] https://doi.org/10.48550/arXiv.2006.15222
>
> _**Question 2:** Given the reliance on specific protein language models (ESM2 and SaProt), have you evaluated the model’s adaptability to other_
>
> **Response:** We thank the reviewer for the valuable question. The purpose of this method is to predict the inter-protein contacts, which can be used for protein docking and structure prediction. In other studies, the representations of PLM are used to predicting the protein binding sites[1-2], protein-protein interactions[3-4] and so on.
>
> [1] https://doi.org/10.1038/s42003-023-04462-5
>
> [2] https://doi.org/10.7554/eLife.93695.3
>
> [3] https://doi.org/10.1016/j.ab.2024.115550
>
> [4] https://doi.org/10.1038/s41598-023-31612-w
>
> _**Question 3:** Have you considered evaluating DeepSSInter on other types of protein complexes, such as oligomeric or multimeric structures? If so, what were the outcomes?_
>
> **Response:** We thank reviewer for the valuable questions/suggestion. DeepSSInter is designed for dimeric complexes, and therefore will only work the best on dimeric complexes, although it still works okay with oligomeric or multimeric structures. One may retrain DeepSSInter on oligomeric or multimeric structures to address the problem.
>
> _**Question 4:** Could you elaborate on the choice of using both ResNet-Inception and triangle-aware modules? How do these components interact, and do they provide overlapping or distinct contributions to model performance?_
>
> **Response:** We thank the reviewer for the valuable question. The purpose of this method is to predict inter-protein contacts. The interacting contacts may be located in long range in the sequence dimension. Therefore, we apply the ResNet-Inception with three different branch of convolution kernels to expand the effective receptive filed. In addition, the triangle-aware module is proposed to consider the multi-body effects and reduce the nonconsistency of geometric triangle inequality. The detailed roles of these two modules may refer to following reference. We have added the corresponding citation in the manuscript (page 4 in the manuscript).
>
> Lin, P., Tao, H., Li, H., & Huang, SY. Protein-protein contact prediction by geometric triangle-aware protein language models. Nature machine intelligence, 5, 1275-1284 (2023).
>
> _**Question 5:** Given the performance decline with AlphaFold2-predicted structures, have you considered fine-tuning the model on such predicted structures? If so, what were the results?_
>
> **Response:** We thank the reviewer for the valuable questions. It is a nice suggestion to fine-turn the model with AlphaFold-predicted structures to improve the robustness and enhancing real-world applicability. However, it is extremely time-consuming to predicted monomer structures with AlphaFold2 in the training set and insufficient to provide results of the fine-tune model within such a short time frame. Moreover, we have seen similar fine-turn or training strategies in other studies (Ref. SaProt), and their experiments have demonstrated that training using AF2-predicted structures does not yield significant improvements.

---

> > ### Comment · Reviewer_xbjr · 2024-11-26
> >
> > I am satisfied with the authors’ responses.

---

> > > ### Author Response · Authors · 2024-11-27
> > > **Response to Reviewer xbjr**
> > >
> > > **Response:** We very appreciate the reviewer’s valuable time for reviewing our revised manuscript. We are also pleased to know that the reviewer is satisfied with our responses, and hope that the reviewer can give a re-rating of our study.

---

### Official Review · Reviewer_YoDi · 2024-11-03

**Soundness:** 2
**Presentation:** 2
**Contribution:** 1
**Rating:** 3
**Confidence:** 5

**Summary:**

The paper introduces DeepSSInter, which predicts protein-protein interactions by combining existing protein language models.

**Strengths:**

evaluation performances are better than selected baselines.

**Weaknesses:**

The paper seems more like a wrapped-up of existing methods and provides little new insights on the problem, making the submission a workshop-level paper instead of research paper.

The paper largely ignores existing methods in multimeric protein structure predictions with MSAs ([1,2]) or PLMs ([3,4]). These methods share identical motivations to the proposed one, especially [4] which proposes to predict protein-protein interactions using protein language models. Discussions and comparisons should be made against these methods. Also comparisons with protein-protein docking methods ([5,6]) should be discussed, as these methods are already broadly used in real applications.

[1] AlphaFold Multimer https://www.biorxiv.org/content/10.1101/2021.10.04.463034v1 ;
[2] RosettaFold https://www.science.org/doi/10.1126/science.abj8754 ;
[3] ESM-Fold https://www.science.org/doi/10.1126/science.ade2574;
[4] Uni-Fold MuSSe: https://www.biorxiv.org/content/10.1101/2023.02.14.528571v1 ;
[5] HDock: https://www.nature.com/articles/s41596-020-0312-x ;
[6] Haddock: https://rascar.science.uu.nl/haddock2.4/ ;

**Questions:**

Figure 3 and 4 should be reorganized  for readers to extract information more easily. Currently it is hard for readers to accurately compare all the colored bars.

---

> ### Author Response · Authors · 2024-11-23
> **Responses to Reviewer YoDi**
>
> We thank the reviewer for reviewing our manuscript and giving the valuable comments. We have carefully addressed the weaknesses/questions and conducted additional evaluations/comparisons.
>
> Our manuscript has been updated accordingly. We really appreciate the reviewer’s valuable time and comments, and hope the reviewer can reevaluate our study by considering our responses below.
>
> _**Weaknesses:**_
>
> _**Weakness 1:** The paper seems more like a wrapped-up of existing methods and provides little new insights on the problem, making the submission a workshop-level paper instead of research paper._
>
> **Response:** We thank the review for the comment. The main goal of this study is to propose a deep learning-based integrative model, and then apply it to solve the inter-protein residue-residue contact prediction problem in the biology field. Apparently, a simple wrapped-up of existing methods will not work. In fact, our model needs to leverage the embedding features of different protein language model. In addition, we also utilize the geometric graph transformer to process the protein structure for extracting the structure representation to improve the accuracy of our network.
> As shown in the significantly better performance of DeepSSInter than other methods, our present study does serve the purpose of the ICLR section “applications in audio, speech, robotics, neuroscience, biology, or any other field”.
>
> _**Weakness 2:** The paper largely ignores existing methods in multimeric protein structure predictions with MSAs ([1,2]) or PLMs ([3,4]). These methods share identical motivations to the proposed one, especially [4] which proposes to predict protein-protein interactions using protein language models. Discussions and comparisons should be made against these methods. Also comparisons with protein-protein docking methods ([5,6]) should be discussed, as these methods are already broadly used in real applications._
>
> **Response:** We thank the reviewer for the valuable comments and suggestions. We have added the corresponding discussions (see page 7 of the manuscript), and calculated the corresponding results of AlphaFold-Multimer, ESMFold, and HDOCKlite (the ab initio docking version of HDOCK) on the Homodimer289 and Heterodimer99 test sets based on the more realistic AlphaFold2-predicted monomer structures. Since the primary goal of current method is to address the situation with limited sequence in MSA. Therefore, we only use the single-sequence and no template for AlphaFold-Multimer. In addition, for the limited time reason, we did not test Uni-Fold MuSSe and HADDOCK at this point.
>
> Please see Tables 2 and 4 for detailed results. As shown in Tables 2 and 4, DeepSSInter also outperforms these methods.
>
> _**Weakness 3:** Figure 3 and 4 should be reorganized for readers to extract information more easily. Currently it is hard for readers to accurately compare all the colored bars._
>
> **Response:** We thank the reviewer for the valuable comment and suggestion. We have reorganized the two figures and merged the same top-n precisions of different ablation models for comparison.
>
> Please see the updated Figures 3 and 4 for the revision.

---

### Official Review · Reviewer_n9PY · 2024-11-04

**Soundness:** 2
**Presentation:** 2
**Contribution:** 2
**Rating:** 3
**Confidence:** 3

**Summary:**

This work proposes DeepSSInter, a transformer-based deep learning model for predicting protein-protein interface contacts without relying on Multiple Sequence Alignments (MSA). DeepSSInter combines single-sequence and structure-aware protein language models, utilizing intra-protein distance and graph representations to capture the structural and evolutionary properties of interacting proteins. In particular, the model leverages ESM2 and SaProt representations, a ResNet-Inception module, and a geometric triangle-aware module to enhance prediction accuracy. Experiments on homo- and heterodimeric complexes demonstrate that DeepSSInter outperforms state-of-the-art methods, particularly in challenging heterodimeric cases.

**Strengths:**

The work's primary strength lies in its approach to protein-protein contact prediction, addressing the limitation of existing methods by eliminating the need for Multiple Sequence Alignments (MSA). The method’s reliance on single-sequence data and its use of structure-aware language models (ESM2 and SaProt) offer computational efficiency and scalability, broadening its applicability to proteins with limited or no homologous sequences.

**Weaknesses:**

The whole work is like a variant of DeepInter [1] with incremental improvements. The architecture of the proposed model only changes the way of extracting embeddings from input when compared to DeepInter. The datasets and experimental settings are all the same as those in DeepInter.

The description of the proposed method is not clear. It is unknown about the details of the "ResNet-Inception module" and "triangle-aware module". It seems that they are directly borrowed from DeepInter, but the authors didn't mention that in the manuscript.

In Table 4, it is misleading that the results of the proposed DeepSSInter are all highlighted but they all perform worse than DeepInter.


[1] Lin, P., Tao, H., Li, H., & Huang, SY. Protein-protein contact prediction by geometric triangle-aware protein language models. Nature machine intelligence, 5, 1275-1284 (2023).

**Questions:**

How the ligand 1D and receptor 1D features are combined with those 2D features?

How the attention representations are generated by ESM2 and SaProt? Each hidden layer of PLMs can have attention representations.

Are there multimers for the receptors in the datasets? If so, how are they processed by the PLMs?

---

> ### Author Response · Authors · 2024-11-23
> **Responses to Reviewer n9PY**
>
> We thank the reviewer for reviewing our manuscript and giving the valuable comments. We have carefully addressed the weaknesses/questions and made the corresponding corrections.
>
> The main goal of this study is to propose a deep learning-based integrative model, and then apply it to solve the inter-protein residue-residue contact prediction problem in the biology field. As shown in the significantly better performance of DeepSSInter than other methods, our present study does serve the purpose of the ICLR section “applications in audio, speech, robotics, neuroscience, biology, or any other field”.
>
> Our manuscript has been updated accordingly. We really appreciate the reviewer’s valuable time and comments, and hope the reviewer can reevaluate our study by considering our responses below.
>
> _**Weaknesses:**_
>
> _**Weakness 1:** The whole work is like a variant of DeepInter [1] with incremental improvements. The architecture of the proposed model only changes the way of extracting embeddings from input when compared to DeepInter. The datasets and experimental settings are all the same as those in DeepInter._
>
> **Response:** We thank the reviewer for the valuable comments. DeepInter requires MSA, which may limit its usage. The primary goal of current method is to address the issue of limited sequences in the MSA of protein complexes. Therefore, the main approach is to leverage the embedding features of single-sequence protein language models. In addition, we also utilize the geometric graph transformer to process the protein structure for extracting the structure representation to improve the accuracy of our network.
>
> _**Weakness 2:** The description of the proposed method is not clear. It is unknown about the details of the "ResNet-Inception module" and "triangle-aware module". It seems that they are directly borrowed from DeepInter, but the authors didn't mention that in the manuscript._
>
> **Response:** Yes, these two modules are directly borrowed from DeepInter. We have added the corresponding citation to indicate its source in the manuscript.
>
> _**Weakness 3:** In Table 4, it is misleading that the results of the proposed DeepSSInter are all highlighted but they all perform worse than DeepInter._
>
> **Response:** We thank the reviewer for pointing out this mis-format. We have now correctly highlighted the best values for each of the evaluated metrics in Table 4.
>
> _**Questions:**_
>
> _**Question 1:** How the ligand 1D and receptor 1D features are combined with those 2D features?_
>
> **Response:** The receptor and ligand 1D features are duplicated across the row and column dimensions and concatenated along the hidden dimension, which forms a 2D feature and combined the other preprocess 2D features. For example, if the shape of receptor and ligand 1D features are (Nrec, H) and (Nlig, H), the shapes of duplicated features will be (Nrec, Nlig, H).
>
> _**Question 2:** How the attention representations are generated by ESM2 and SaProt? Each hidden layer of PLMs can have attention representations._
>
> **Response:** For the ESM2 and SaProt models, we provide the required input features and extract the hidden attention features of all layers, which are concatenated along feature dimension. Specifically, the ESM2 and SaProt both have 33 layers and 20 multi-head for each layer. Therefore, the attention representation of the two models are (Nrec, Nlig,660).
>
> _**Question 3:** Are there multimers for the receptors in the datasets? If so, how are they processed by the PLMs?_
>
> **Response:** We thank the reviewer for the question. In current work, our datasets only include the protein dimers. However, it is easily to extend the current model to address the receptors with multiple chains. For extracting the representation of PLM, it is just to link the corresponding sequences for ESM2, and the foldseek tokens of structures for SaProt.

---

### Meta-Review · Area_Chair_8MZa · 2024-12-21

**Metareview:**

(a) The paper proposes DeepSSInter, a transformer-based deep learning model designed to predict inter-protein residue-residue contacts by integrating single-sequence and structure-aware protein language models (PLMs). It aims to improve the prediction of interface contacts between interacting proteins without relying on Multiple Sequence Alignments (MSA), which can limit prediction accuracy, speed, and computational efficiency.

(b) Strengths of the paper include its innovative attempt to move beyond MSA for predicting protein interactions, thereby addressing challenges in computational complexity and prediction accuracy. The model's design incorporates advanced components like graph-based features, ResNet-Inception, and a triangle-aware module, which are intended to enhance performance. The authors also provide experimental validation across different datasets, demonstrating improvements over several baselines.

(c) Weaknesses and missing elements
- Lack of novelty: The work appears to be a variant of DeepInter with incremental changes. Key modules like the "ResNet-Inception module" and "triangle-aware module" are directly borrowed, and the datasets and experimental settings are similar.
- Initial method description: The original description of the proposed method was unclear, particularly regarding the details of the borrowed modules.
- Performance with AlphaFold2-predicted structures: The model's performance declines with AlphaFold2-predicted structures, especially for heterodimers, indicating potential limitations in handling complex structures and suggesting a lack of coevolutionary information compared to MSA-based models.
- Interpretability: The interpretability of the model's outputs is not addressed, which is crucial for practical applications in understanding the basis of contact predictions.
- Testing scope: Testing is mainly focused on homo- and heterodimeric complexes, and could be broadened to include more diverse protein types like oligomeric or multimeric complexes.

(d) The most important reasons for rejecting this paper are the limited novelty in the proposed architecture compared to existing models, inconsistent performance improvements over baseline methods, and insufficient exploration of the model's limitations and generalizability. These factors raise concerns about the paper's contribution to advancing the field.

**Additional Comments On Reviewer Discussion:**

During the rebuttal period, reviewers raised several points regarding the originality, comparative analysis, and robustness of DeepSSInter. Reviewers questioned the extent of innovation given similarities with DeepInter and highlighted inconsistencies in performance gains. Authors acknowledged borrowing components from DeepInter and clarified that their main contribution lies in leveraging PLMs for cases with limited MSA data. However, they did not adequately address why their method should be preferred over DeepInter or how it fundamentally advances the field beyond incremental improvements.

The authors' responses, while showing some effort, did not fully overcome the concerns raised by the reviewers. The lack of significant novelty, unresolved issues with model performance and limitations, and the need for further improvements in clarity and comprehensiveness led to the decision to reject. The additional evaluations and explanations did not convincingly demonstrate that the paper had reached a level worthy of acceptance.

---

### Decision · Program_Chairs · 2025-01-22

Reject